# Boosting the Performance of Perovskite Solar Cells through Systematic Investigation of the Annealing Effect of E-Beam Evaporated TiO_2_

**DOI:** 10.3390/mi14061095

**Published:** 2023-05-23

**Authors:** Tao Xue, Dandan Chen, Ting Li, Xingxing Chou, Xiao Wang, Zhenyu Tang, Fanghui Zhang, Jin Huang, Kunping Guo, Ashkan Vakilipour Takaloo

**Affiliations:** 1School of Electronic Information and Artificial Intelligence, Shaanxi University of Science and Technology, Xi’an 710021, China; 2School of Electrical and Control Engineering, Shaanxi University of Science and Technology, Xi’an 710021, China; 3State Key Laboratory for Artificial Microstructures and Mesoscopic Physics, Department of Physics, Peking University, Beijing 100871, China; 4Department of Chemical and Process Engineering, University of Canterbury, Christchurch 8140, New Zealand

**Keywords:** TiO_2_ electron transport layer, fill factor, CsPbI_3-*x*_Br*_x_* perovskite solar cell

## Abstract

Electron transport layer (ETL) plays an undeniable role in improving the performance of n-i-p planar perovskite solar cells (PSCs). Titanium dioxide (TiO_2_) is known as a promising ETL material for perovskite solar cell. In this work, the effect of annealing temperature on optical, electrical, and surface morphology of the electron-beam (EB)-evaporated TiO_2_ ETL, and consequently on the performance of perovskite solar cell, was investigated. It was found that annealing treatment at an optimized temperature of 480 °C considerably improved the surface smoothness, density of grain boundaries, and carrier mobility of TiO_2_ film, which resulted in nearly 10-fold improvement in power conversion efficiency (11.16%) in comparison with the unannealed device (1.08%). The improvement in performance of the optimized PSC is attributed to the acceleration of charge carrier extraction, as well as suppression of the recombination at the ETL/Perovskite interface.

## 1. Introduction

Organic–inorganic hybrid perovskite solar cells (PSCs) have attracted considerable attention as one of the emerging research trends for both the academic and industrial sectors in the past decade. PSCs offer intriguing optoelectronics properties such as long carrier life time, high absorption coefficient, intrinsic low trap density, and ambipolar transport properties [1,2,3,4,5,6]. According to the National Renewable Energy Laboratory (NREL) chart, PSCs have demonstrated rapid progress in the performance enhancement from 3.8% [7] to a certified 25.7% [8]. Among the different components of PSC, the electron transport layer (ETL) plays a critical role in extracting and transporting photogenerated electron carriers and suppressing charge recombination in photovoltaic devices [9]. Therefore, interface engineering is now regarded as an efficient strategy to improve the power conversion efficiency (PCE) of perovskite solar cell devices by preventing the undesired degradation pathways [10]. This is mainly due to the fact that the photoinduced carriers not only have to transport across the interfaces in the cell but also charge extraction generally occurs at the interfaces, thereby leading to undesirable charge losses due to any possible interfacial defects and the associated specific charge distributions [10,11].

Recently, intensive efforts have been devoted to improving the electron transport layer (ETL)/perovskite layer, perovskite layer/hole transport layer, and the front and back contact interfaces [12,13,14]. Concurrently, various functional materials have been investigated to overcome the interfacial losses in PSC for performance enhancement. One of the most attractive target materials is titanium dioxide (TiO_2_) due to its appealing advantages, such as high transparency, favorable bandgap edge positions in relation to perovskites, environmental stability, etc. For instance, Li et al. successfully grew an array of anatase-phase TiO_2_ nanorods on FTO conductive glass using a hydrothermal method, enabling a 15.3% efficiency for PSC [15]. Tian et al. fabricated a composite TiO_2_ ETL using a mixed spray pyrolysis method, demonstrating that TiO_2_ nanoparticles can promote the charge extraction process at the TiO_2_/perovskite interfaces and improve the recombination process in perovskite layers [16]. Very recently, magnetron sputtering technology on FTO glass was also used to fabricate different thicknesses and shapes of TiO_2_ ETL for PSC [17,18]. Electron beam (EB) evaporation of TiO_2_ has emerged as a promising and cost-effective approach for large-scale production of perovskite solar cells (PSCs) [19]. The annealing process for TiO_2_ probably plays a key role in the crystalline perovskite phase and subsequently the charge extraction process. However, previous studies have primarily focused on PSCs utilizing indium tin oxide (ITO) substrates and organic–inorganic perovskite layers [20]. Considering the scarcity of indium, it is of greater practical significance to explore PSCs based on fluorine-doped tin oxide (FTO) substrates. Moreover, inorganic perovskite materials theoretically exhibit superior stability compared to their organic counterparts. Consequently, investigating the impact of annealing on the characteristics of perovskite solar cells prepared via EB evaporation of TiO_2_ on FTO substrates holds substantial importance.

In this context, we have investigated the effect of annealing temperature on the optical, electrical, and morphological properties of EB-evaporated TiO_2_ films to be used in PSC devices. Upon varying the annealing temperature, we demonstrated the temperature-dependent morphology and opto-electronic properties of TiO_2_ as a ETL for PSC. The 480 °C-heat-treated TiO_2_ film led to the formation of a uniform and dense surface of TiO_2_ film with minimum defects and high carrier mobility. Notably, the PSC with TiO_2_ films annealed at 480 °C showed a significant increase in the fill factor (73%), which is 3.5 times higher than that of an unannealed device (16%). For the power conversion efficiency, an approximately 10-fold improvement (11.16%) for 480 °C annealed PSC was obtained as compared to the reference device (1.08%). To unveil the internal conversion mechanism, a detailed analysis of electrical properties and charge transfer at the TiO_2_/perovskite interface was also performed.

## 2. Materials and Methods

### 2.1. Fabrication of CsPbI_3-x_Br_x_ Perovskite Solar Cells

TiO_2_ thin films were deposited on fluorine-doped tin-oxide-coated glass (FTO/glass) by EB evaporation, as shown in step 1 of Figure 1. Before deposition, the chamber was evacuated to a base pressure of 5 × 10^−3^ Pa and the substrate was heated to 350 °C. TiO_2_ particles were used as the source for evaporation, the deposition rate and film thickness were set at 0.5 Å·s^−1^ and 75 nm, respectively, and oxygen was introduced during the deposition process while the pressure was kept at 3.3 × 10^−2^ Pa. Annealing of the TiO_2_ thin film was carried out in oxygen at 200 °C, 300 °C, 400 °C, 450 °C, 480 °C, and 500 °C for 2 h using a tube furnace, as shown in step 2 of Figure 1. A mixture of 380.4 mg HPbI_3_, 187.08 mg CsI, and 73.4 mg PbBr_2_ was dissolved in DMF and DMSO and filtered with a 0.2 μm syringe filter to form clear CsPbI_3-*x*_Br*_x_* solution. CsPbI_3-*x*_Br*_x_* films were casted by spin coating in a glove box at 4500 r.p.m for 30 s and dried on a hot plate at 200 °C for 5 min, as shown in step 3 of Figure 1. HTL was applied immediately after the perovskite film was cooled. The precursor solution for the HTL was prepared by dissolving 72.3 mg spiro-OMeTAD, 29 μL TBP, and 18 μL Li-TFSI solution (520 mg Li-TFSI in 1 mL acetonitrile) in 1 mL CB. The HTL was casted via spin coating at 7050 r.p.m. for 30 s in the glove box, as shown in step 4 of Figure 1. The samples were then stored in a dark desiccator for the night. A 70 nm thick Ag layer was then deposited on top of the HTL through a metal shadow mask by thermal evaporation at a base pressure of about 2 × 10^−3^ Pa, as shown in step 5 of Figure 1.

### 2.2. Characterization

The surface morphologies of the samples were examined using a Field Emission Scanning Electron Microscope (FESEM, S-4800, Hitachi, Tokyo, Japan) and scanning probe microscope (Dimension edge, Bruker, Saarbrucken, Germany). Current density–voltage (*J-V*) curves were measured in dry air using a Keithley 2400 source meter under standard 1 sun AM 1.5 simulated solar irradiation system (SAN-EI 100 mw·cm^−2^) from Giant Force Technology Co., Ltd., New Taipei City, Taiwan. The scan rate of 50 mV·s^−1^ was adopted in the *J-V* measurement. The absorption properties of films were determined using the UV-Vis spectrophotometer (Lambda 850, Perkin Elmer, Waltham, MA, USA). Photoluminescence (PL) spectra were measured by using an Edinburgh, FLS 980 fluorescence spectrometer under excitation at 375 nm.

## 3. Results

### 3.1. Morphology Properties of TiO_2_ Films

In order to evaluate the surface morphology of the TiO_2_ film, Atomic Force Microscopy (AFM) (SPI3800N/SPA400, Rigaku, Tokyo, Japan) was employed, and Scanning Electron Microscopy (SEM) (S-4800, Hitachi, Tokyo, Japan) analyses were carried out on unannealed and annealed samples with different annealing temperatures ranging from 200 °C to 500 °C. As shown in Figure 2(A1–A7), the surface roughness of the unannealed TiO_2_ film was 33.91 nm. As the TiO_2_ film underwent annealing treatment, the roughness decreased from 32.10 nm to 26.17 nm with an increase in annealing temperature from 200 °C to 480 °C, respectively. Surprisingly, as the annealing temperature rose from 480 °C to 500 °C, the roughness remained constant and is expected to increase further for higher temperatures. Figure 2(B1–B7) represents the top-view SEM characterization of unannealed and annealed TiO_2_ films. It was observed that the surface of TiO_2_ contained a combination of large (0.325–0.75 μm) and small grains (0.025–0.3 μm), along with some pinholes and voids in the range of 0.025 μm to 0.1 μm.

Statistical data Figure 2(C1–C7) reveals that as the annealing temperature escalated from 200 °C to 480 °C, the percentage of voids dramatically declined from 4% to 0.7%. However, as the temperature rose to 500 °C, the number of pinholes and voids significantly increased to 2%, as shown in the red indicated area in Figure 2(B1–B7). Therefore, it is assumed that 480 °C is an optimized annealing temperature resulting in the formation of TiO_2_ with a smooth surface with a minimum number of voids. This high-quality film is favorable for the development of crystals with regular shapes [21].

Figure 3 provides a schematic illustration of the reason behind the aforementioned phenomena. It is observed that prior to annealing treatment, the evaporated TiO_2_ film consists of tiny grains and a large number of surface defects comprising pinholes and small voids. Table 1 displays a proportion between voids and grain sizes as a function of annealing temperature. In the low-temperature mode, ranging from 200 °C to 400 °C, with increasing the annealing temperature, the number of small-sized grains outnumbered large-sized grains. At the same time, the number of voids steady declined, which is assumed to be due to the gradually growing grains. Therefore, at 400 °C, the highest proportion of small-sized grains (88.00%) to large-sized grains (10.02%) was formed. In contrast, as the annealing temperature exceeded 400 °C, enough energy was provided for large grains to absorb nearby small-sized grains and become thermodynamically stable through lowering the surface free energy [22]. Therefore, in the second temperature mode, above 450 °C, the proportion of large-sized grains to small-sized grains was reversed, and at 500 °C, the highest number of large-size grains (21.8%) to small-sized grains (76.0%) was formed [22,23]. Moreover, the number of voids experienced a decreasing trend from 200 °C (4%) to 480 °C (0.7%) and rose again as annealing temperature exceeded 480 °C. The reason behind this phenomenon is associated with the difference between the surface tensions and thermal expansion coefficients of the various grains in the film [24]. Therefore, the grains started to break up and more voids occurred as the annealing temperature rose to 500 °C.

### 3.2. Photovoltaic Performance of CsPbI_3-x_Br_x_ Perovskite Solar Cells

To evaluate the effect of heat-treated TiO_2_ film as an electron transport layer on the performance of a solar cell device, TiO_2_ films with different annealing temperature were implemented in CsPbI_3-*x*_Br_*x*_ PSCs. Figure 4a displays the current density–voltage (*J-V*) characteristics of PSC based on TiO_2_ ETLs with different annealing temperature and Table 2 shows the photovoltaic parameters of corresponding PSCs with various TiO_2_ ETLs. The device based on unannealed TiO_2_, without annealing treatment, exhibited a short circuit current density (*J*_sc_) of 8.76 mA·cm^−2^, an open-circuit voltage (*V*_oc_) of 0.76 V, a fill factor (FF) of 16%, and a PCE of 1.08%. After TiO_2_ annealing at 200 °C, there was no change in the *V*_oc_ (0.76 V); however, *J*_sc_ was increased to 12.96 mA·cm^−2^, FF was increased to 22%, and finally a PCE of 2.15% was achieved. This increasing trend in PSC’s photovoltaic parameters were continued as the annealing temperature was enhanced to 480 °C, where a *V*_oc_ of 0.84 V, a *J*_sc_ of 18.33 mA·cm^−2^, a FF of 73%, and eventually a PCE of 11.16% were obtained. Surprisingly, as the annealing temperature of TiO_2_ was increased from 480 °C to 500 °C, the photovoltaic parameters of the PSC began to decline. Therefore, PSC with TiO_2_ annealed at 480 °C showed the outstanding performance among other candidates. To explore the causes behind the performance improvement of PSCs possessing TiO_2_ film with different annealing temperatures, the resistivity of unannealed and annealed TiO_2_ at various temperature ranging from 200 °C to 500 °C was measured as represented in Figure 4b. As is shown, the resistivity of unannealed TiO_2_ film was 1.34 Ω·m, and as the annealing temperature raised from 200 °C to 480 °C, the resistivity of TiO_2_ film dropped from 1.30 Ω·m to 0.10 Ω·m, respectively, and then increased to 0.24 Ω·m as the annealing temperature approached 500 °C. The series resistance of PSC is affected by several factors, including the resistivity of charge transport layers. Moreover, it is reported that the internal series resistance of the solar cell device tends to lower the fill factor substantially [25], hence limiting the overall performance of solar cell devices [26,27].

The power lost through the solar cell can be expressed as
$$P_{\mathrm{loss}}={\left(I_{sc}^{*}\right)}^{2}\times R_{S}$$
where *P*_loss_ is power loss, *I*_sc_ is short circuit current, and *R*_s_ is internal series resistance [28]. The power loss increases with the increasing series resistance, and hence there is a drop in the fill factor as seen in Table 2. Therefore, as the resistivity of TiO_2_ film is increased, as a function of annealing temperature, the series resistance is enhanced, which deteriorates the performance of the device as shown in Table 2.

Figure 5a schematically illustrates the mechanism behind the movement of carriers in EB-evaporated TiO_2_ film, hence affecting the performance of PSC. Essentially, TiO_2_ film consists of grain, grain boundaries, and numerous defects which play a role as barriers hindering the movement of photogenerated electrons. These crystal defects scatter the conduction of electrons and impede the flow of current leading to performance degradation of PSC [29].

Elemental composition analysis of the TiO_2_ film was performed using Energy-Dispersive X-ray Spectroscopy (EDS) (S-4800, Hitachi, Tokyo, Japan). Figure 5b shows the EDS spectra of the surface of TiO_2_ film annealed at 480 °C (for more details, please see Figure A1 in Appendix A). It is revealed that the surface of TiO_2_ film annealed at 480 °C contains less imperfections with the minimum ratio of titanium to oxygen 1:2. This minimum amount of imperfections provides less interfacial obstacles in the way of electron transition in crystal structure of the ETL TiO_2_ film [30].

The statistical analysis of the photovoltaic device involved the use of 20 devices. Among the different annealing temperatures, the TiO_2_ ETLs film annealed at 480 °C exhibited favorable characteristics, such as good surface coverage and smooth surface without gaps. These attributes contributed to the reduction in defect state density and the accumulation of interface charges, resulting in a decreased hysteresis effect (for more details, please see Table A1 and Figure A2 in Appendix A). To evaluate the long-term stability of the devices, they were stored and tested in ambient air humidity conditions at room temperature within the range of (20 ± 5) °C and RH = 20 ± 5%, without encapsulation. The findings reveal that the PSCs fabricated with TiO2 ETLs annealed at 480 °C exhibited notable advantages in terms of reliability, reproducibility, and overall performance (for more details, please see Figure A3 in Appendix A).

### 3.3. Opto-Physical Properties

Optical transparency of ETL is a critical factor determining the performance of n-i-p-structured PSC since it allows the incoming photons to pass though it and reach to the perovskite layer. Figure 6a shows the transmission spectra of TiO_2_ film with different annealing temperature in the wavelength range of 380 nm to 780 nm. Results show that all annealed samples exhibited an excellent optical transmittance with an average transmittance of above 85% in visible range of wavelength. The small change in optical transparency of TiO_2_ samples with constant thickness (75 nm) is assumed to be due to the difference in refractive index of samples, which is mostly associated with the grain size of TiO_2_ films [31]. We determined the optical band gap (Eg) of TiO_2_ films through the first derivative of transmissivity (T) in relation to energy (E) [32]. Figure 6b shows the effect of annealing temperature on the optical bandgap energy of the TiO_2_ film. It is evident that there is a slight difference in the optical bandgap of TiO_2_ film as a function of different annealing conditions. This result demonstrates that there is a negligible effect of annealing temperature on the optical characteristic of TiO_2_ film. Different annealing conditions easily cause changes in the structure and composition of TiO_2_ films, which in turn can affect their electronic structure and the energy required to excite an electron. For example, annealing at high temperatures can induce crystallization and grain growth, leading to changes in the crystal structure and defects within the film. Consequently, the annealing-treated TiO_2_ films result in a shift in the optical bandgap towards higher or lower energies. In general, the bandgap of a material decreases as the grain size increases, which are attributed to, the presence of grain boundaries in a polycrystalline material induce the defect states in the bandgap, which can act as trapping sites for electrons and holes. These defect states can then lead to a broadening of the bandgap, resulting in a decrease in the optical bandgap of the material. The decrease in bandgap with increasing grain size can be described by the quantum confinement effect, where the electronic states in the material are confined by the grain boundaries, leading to a change in the electronic band structure. The degree of quantum confinement is related to the grain size, with smaller grains resulting in a greater degree of confinement and a larger increase in bandgap. However, it is important to note that the relationship between grain size and bandgap can also be influenced by other factors, such as strain and doping. In some cases, an increase in grain size can lead to a decrease in strain, which can in turn lead to a decrease in the bandgap. Additionally, doping can introduce new electronic states into the bandgap, which can affect the bandgap energy and modify the relationship between grain size and bandgap.

To evaluate the carrier dynamics at the interface between the absorber layer (CsPbI_3-*x*_Br_*x*_ films) and the ELT layer (TiO_2_ film) with different annealing temperatures, photoluminescence (PL) spectra analysis was carried out in the wavelength range of 550 nm to 850 nm. Figure 7 shows the steady-state PL spectra of CsPbI_3-*x*_Br*_x_* perovskite film coated with TiO_2_ with different annealing temperatures. As is observed, the PL luminescence intensity is maximized for TiO_2_ film without annealing treatment; however, as the annealing temperature is enhanced, the intensity of the PL peak is quenched and reaches its minimum value at 480 °C, indicating more effective electron extraction. The PL intensity is raised again when the annealing temperature reaches 500 °C. As is known, defects such as voids, grain boundaries, and vacancies that form at the interface between the TiO_2_ ETL and perovskite film can contribute to recombination loss [33]. Thus, TiO_2_ film annealed at 480 °C significantly minimizes the number of these trap states and consequently enhances the acceleration of the carrier’s extraction from the perovskite layer. As the annealing temperature approaches 500 °C, this tendency is broken and the PL intensity is raised again due to an increase in the number of recombination sites, such as defects and voids, as shown in Table 1.

The space-charge-limited-current (SCLC) test was conducted using the structure shown in Figure 8a, and the results are presented in Figure 8b, providing valuable insights into the recombination behavior at the interface between the ETL and the perovskite material. This analysis offers an additional perspective that aids in understanding the observed trend in the open-circuit voltage (*V*_oc_). SCLC is a useful technique for investigating carrier mobility and recombination properties in materials. In SCLC, when trap centers capture photogenerated free carriers, the density of free carriers decreases, resulting in a small current with a shallow slope at low bias voltage. However, as the voltage increases, all traps become filled. Beyond the inflection point, the current increases rapidly with a steeper slope, and this inflection point is known as the trap-filled limit voltage (*V*_TFL_). The value of *V*_TFL_ can be estimated by calculating the intersection point of the two slopes. Additionally, *V*_TFL_ can be determined by the trap state density (*N*_t_). The values of *N*_t_ for the TiO_2_ films before and after annealing at 200 °C, 300 °C, 400 °C, 450 °C, 480 °C, and 500 °C were calculated using the formula $V_{\mathrm{TFL}}=\frac{eN_{t}d^{2}}{2\varepsilon \varepsilon_{0}}$, resulting in 8.86 × 10^19^ cm^−3^, 7.67 × 10^19^ cm^−3^, 6.38 × 10^19^ cm^−3^, 6.27 × 10^19^ cm^−3^, 4.10 × 10^19^ cm^−3^, 3.13 × 10^19^ cm^−3^, and 4.10 × 10^19^ cm^−3^, respectively. These results indicate that the increase in *J*_sc_ is primarily attributed to the reduction in trap state density and the decrease in resistivity, as mentioned in this study. Moreover, the decrease in defect state density suggests that the TiO_2_ film annealed at 480 °C exhibits better surface quality, with fewer ETL/perovskite interface recombination sites and defect centers, thereby minimizing the probability of interface recombination.

## 4. Conclusions

In summary, we systematically investigated the effect of annealing temperature on electrical and morphological characteristics of EB-evaporated TiO_2_ thin film as ETL to improve the performance of n-i-p planar PSC. It was found that as the annealing temperature increased from 200 °C to 480 °C, the electrical and morphological properties of TiO_2_ film were improved, which resulted in higher performance of PSC. However, above 480 °C, the quality of TiO_2_ film was deteriorated, which led to lower performance of PSC. It was revealed that the TiO_2_ film annealed at 480 °C contained the lowest amount of surface defects and trap states, which resulted in minimum recombination loss of photogenerated carries. Transmission tests showed an average optical transmittance of above 85% in the visible range of wavelength for the TiO_2_ film annealed at 480 °C, and it was determined that the annealing temperature did not have any influence on the band gap of the TiO_2_ film. It was concluded that annealing treatment at 480 °C remarkably improved the quality of TiO_2_ ETL, resulting in a 3.5-fold improvement in FF and 10-fold increase in device PCE compared to an unannealed device. Our findings provide a straightforward method for improving the FF of perovskite solar cells at a low cost and on a large scale, establishing the groundwork for the commercialization of perovskite solar cells.

## Figures and Tables

**Figure 1 micromachines-14-01095-f001:**
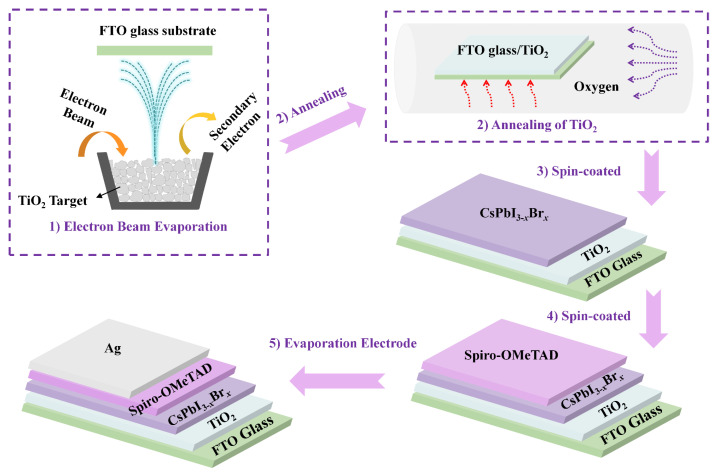
Schematic illustration of the fabrication process for CsPbI_3-*x*_Br*_x_* perovskite solar cell.

**Figure 2 micromachines-14-01095-f002:**
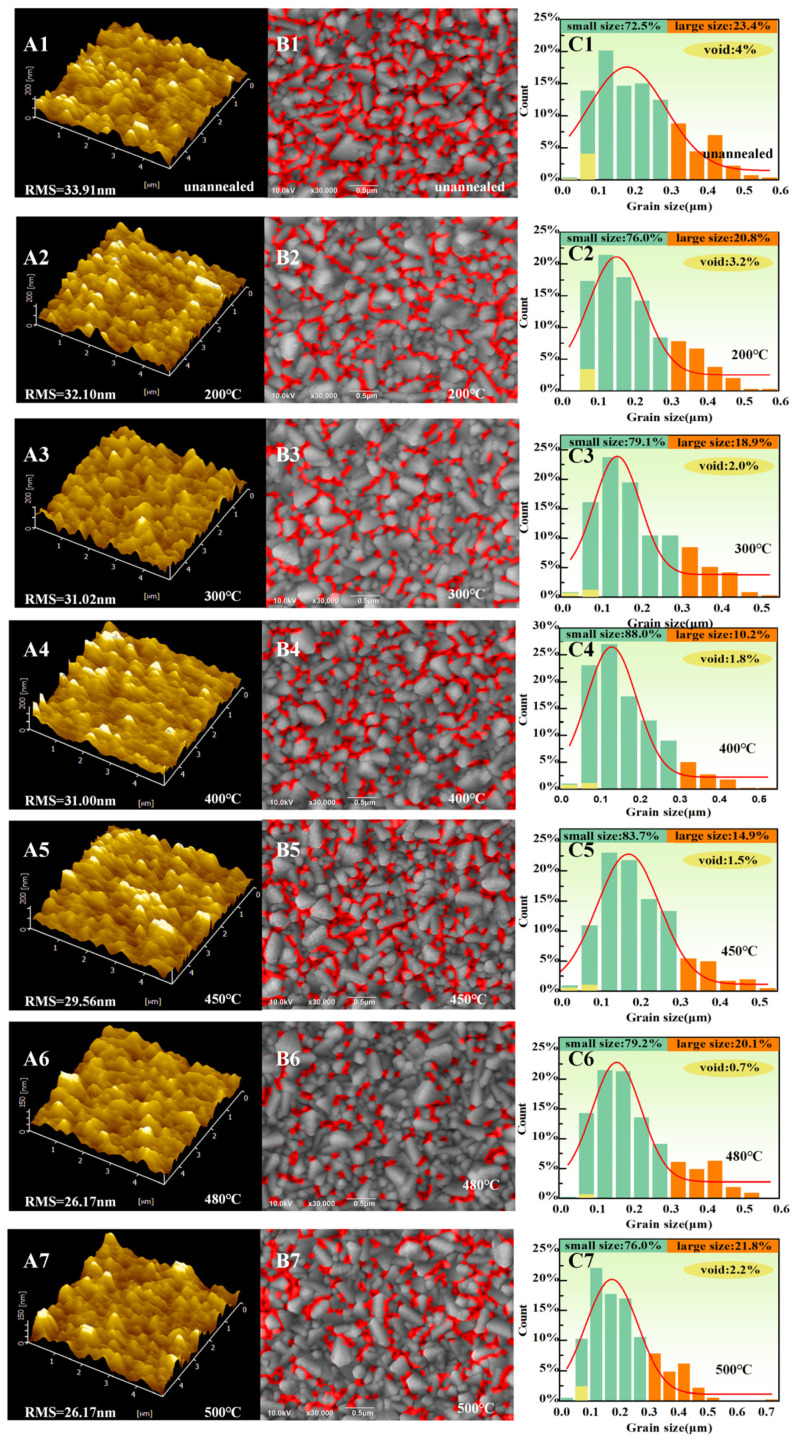
(**A1**–**A7**) AFM image of TiO_2_ films at different annealing temperatures. (**B1**–**B7**) SEM images of the TiO_2_ film at different annealing temperatures. (**C1**–**C7**) Statistics on the percentage of grains and voids of TiO_2_ film at different annealing temperatures.

**Figure 3 micromachines-14-01095-f003:**
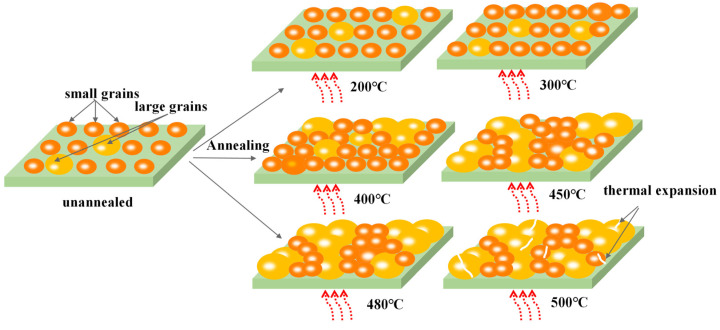
The schematic diagram of TiO_2_ grain growth at different annealing temperatures.

**Figure 4 micromachines-14-01095-f004:**
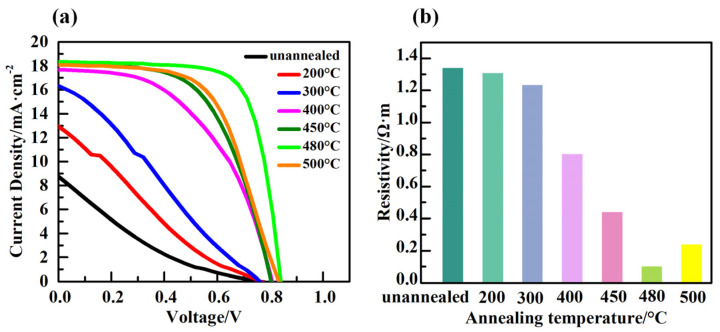
(**a**) *J-V* curves of annealing at different temperatures. (**b**) Resistivity of TiO_2_ films at different annealing temperatures.

**Figure 5 micromachines-14-01095-f005:**
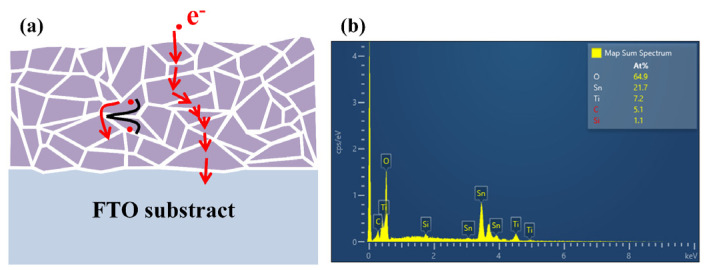
(**a**) Schematic diagram of the internal grain boundaries of TiO_2_. (**b**) EDS pattern of TiO_2_ thin film.

**Figure 6 micromachines-14-01095-f006:**
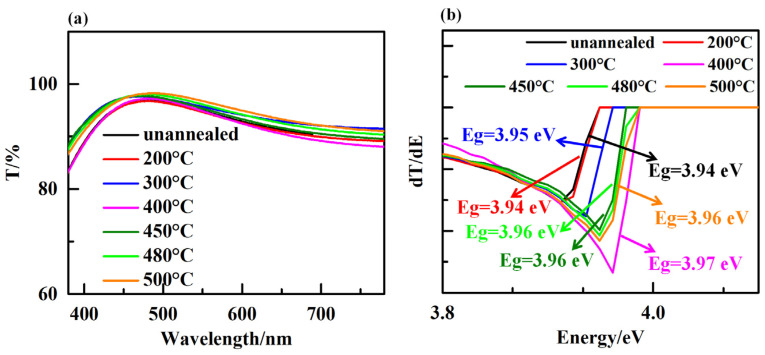
(**a**) Transmittance spectra images of the TiO_2_ film. (**b**) Plots of the derivative of the transmittance with respect to energy.

**Figure 7 micromachines-14-01095-f007:**
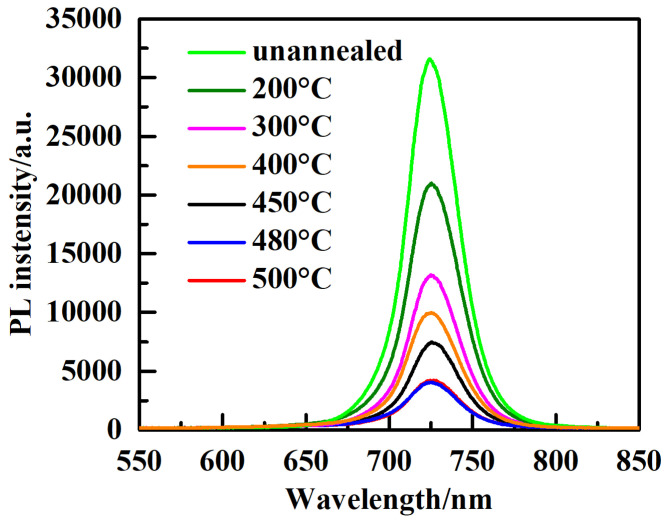
PL spectra of perovskite films prepared with TiO_2_ ETL annealed at different temperatures.

**Figure 8 micromachines-14-01095-f008:**
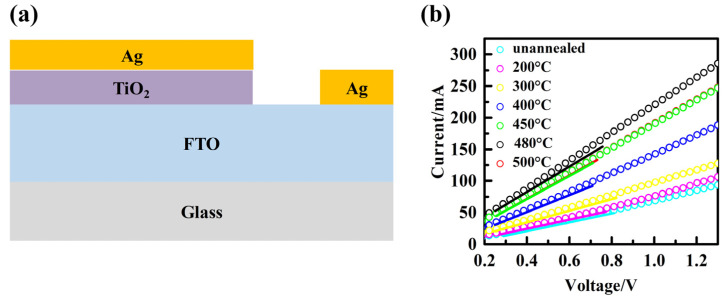
(**a**) Schematic illustration of devices with FTO/TiO_2_/Ag. (**b**) Current–voltage characteristics of devices with FTO/TiO_2_/Ag.

**Table 1 micromachines-14-01095-t001:** Percentage of small grains, large grains, and voids on the surface of the SEM image of the TiO_2_ film.

T/°C	Unannealed	200	300	400	450	480	500
small-size grains	72.5%	76.0%	79.1%	88.0%	83.7%	79.2%	76.0%
large-size grains	23.4%	20.8%	18.9%	10.2%	14.9%	20.1%	21.8%
voids	4%	3.2%	2.0%	1.8%	1.5%	0.7%	2.2%

**Table 2 micromachines-14-01095-t002:** Photoelectric parameters of PSCs with TiO_2_ ETLs annealed at different temperatures.

T/°C	*V*_oc_/V	*J*_sc_/mA·cm^−2^	FF/%	*R*_s_/Ω	PCE/%
unannealed	0.76	8.76	16	2296.41	1.08
200	0.76	12.96	22	661.64	2.15
300	0.76	16.33	27	554.71	3.32
400	0.80	17.66	50	120.81	7.08
450	0.80	18.10	58	104.13	8.43
480	0.84	18.33	73	53.79	11.16
500	0.83	18.07	59	162.90	9.00

## Data Availability

Not applicable.

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
