# Peer review of "Boosting the Performance of Perovskite Solar Cells through Systematic Investigation of the Annealing Effect of E-Beam Evaporated TiO2"

_micromachines, 2023, doi:10.3390/mi14061095_

Round 1
Reviewer 1 Report
I have reviewed the manuscript titled Boosting the Performance of Perovskite Solar Cells through Systematic Investigation of the Annealing Effect of E-beam Evaporated TiO2, and I appreciate the effort and time you have put into this study. Your work on investigating the effect of annealing temperature on the performance of perovskite solar cells and the role of the electron transport layer (ETL) is of great interest.
While the investigation of the effect of annealing temperature on the performance of perovskite solar cells is a significant contribution to the field, the manuscript requires additional experimental results to support the claims made in the study. Specifically, I recommend including more detailed information on the device fabrication process, a comprehensive analysis of the photovoltaic characteristics, and the measurement of additional parameters that would provide a better understanding of the behavior of the devices. Overall, I recommend that you address these issues below before resubmitting your manuscript to Micromachines.
1. Can the author clarify the conditions of "Pristine" clearly? If it is without annealing, I think using "without annealing" or “unannealed” would be clearer than "pristine." And the author needs to unify the usage in the manuscript.
2. What software and method does the author use to determine the grain size and void of SEM, and what are the criteria for distinguishing small size grains from large size grains?
3. In Figure 5, the author uses EDS to explain the titanium to oxygen ratio of TiO2 films annealed at 480°C. Please also provide the titanium to oxygen ratios for the other annealing conditions to reinforce this point.
4. Does the author have cross-section SEM images to demonstrate that TiO2 samples with constant thickness are 75 nm?
5. The author should provide a more in-depth explanation for the slight difference in the optical bandgap of TiO2 films as a function of different annealing conditions.
6. There are huge different Jsc of devices with different annealing temp. The authors should also provide the EQE spectra to explain this and its relationship with the TiO2 film with different annealing temp.
7. The authors may provide the results of space-charge-limited-current (SCLC) or transient photovoltage decay to discuss the recombination behavior at the ETL/perovskite interface to give more point of view to explain the trend of Voc.
8. In Table 2, how many devices were used for the statistical analysis of the photovoltaic device, and please also provide the standard deviation.
9. The authors should provide XRD results to explain the crystallinity of the TiO2 film with different annealing temp. to give another support for the resistivity trend and the electron movement mechanism in Fig 5(a). please also calculate the grain size form the FWHM of XRD peak.
Reviewer 2 Report
Comments to the manuscript ID: micromachines-2400246.
Title: Boosting the Performance of Perovskite Solar Cells through Systematic
Investigation of the Annealing Effect of E-beam Evaporated TiO2
Authors: Tao Xue *, Dandan Chen, Ting Li, Xingxing Chou, Xiao Wang, Zhenyu
Tang, Fanghui Zhang, Kunping Guo, Ashkan Vakilipour Takaloo, Jin Huang
This manuscript studies the effect of annealing temperature on the optical, electrical, and morphological properties of E-beam evaporated TiO2 films to be used in perovskite solar cells (PSCs). The topic of PSCs is of strong scientific interest and this study shows well correlated experimental results that contribute to the improvement performance of these devices. Nevertheless, some minor details need to be clarified before accepting the manuscript.
Introduction.
_ Page 2, lines 58-63. Authors justify the use of TiO2 films for PSCs and comment that only few studies have reported TiO2 films deposited by electron beam evaporation (E-beam). Authors need to add some references. In fact, there exist a study focused on the Annealing effect of E-beam evaporated TiO2 films and their performance in perovskite solar cells (https://doi.org/10.1016/j.jphotochem.2018.04.025). Authors need to compare their results with those reported in literature.
Materials and Methods.
Oxygen based atmosphere in thermal annealing process produce the oxidation of thin films. Why the TiO2 film was thermally annealed in O ambient and not inert atmosphere like nitrogen or argon?
Results.
_ Page 7. Lines 203-204. Regarding the EDS analysis, Authors say: “that the surface of TiO2 film annealed at 480°C contains less imperfections with the minimum ratio of titanium to oxygen, 1:2”. However, they only show the EDS analysis of only 1 sample. It is necessary the analysis of the reference sample for correct comparison. Also, how Authors calculate the Ti:O ratio of 1:2? if the EDS shows 7.2% for Ti and 64.9% for O (Fig. 5b). X-ray photoelectron spectroscopy (XPS) measurements of the films could be of help for a correct compositional analysis.
_ As Authors show, the thermal annealing process affect the structure, optical and electrical properties of the materials. Then, how the thermal annealing affect the electrical properties of FTO bottom electrode?
_ Authors analyze the morphological and the bandgap properties of TiO2 films. Could authors explain the relation between the grain size with the bandgap?
_ The current density vs voltage curves of the different PSCs are shown in figure 4(a). The best performance is obtained with the TiO2 annealed at 480ºC. However, what about the stability and endurance? How is the PCE vs time? The analysis of J-V hysteresis is also important in this kind of devices. Authors need to add these properties to the manuscript.
General comments.
_ Page 2, line 59. Define "EB evaporation".
Correct grammatical mistakes in:
_ Page 3, line 115. “…, the roughness is decreases”.
_ Page 6, line 172. “… parameters began to declined”.
_ Page 7. line 221. “… optical bad gap energy”.
_ Page 8. Line 230. “… the PL spectra analysis was carried out”.
_ Page 8. Line 238. ¿“attribute” or “contribute”? in the sentence: “… perovskite film could attribute to recombination loss”.
Correct minor grammatical mistakes in:
_ Page 3, line 115. “…, the roughness is decreases”.
_ Page 6, line 172. “… parameters began to declined”.
_ Page 7. line 221. “… optical bad gap energy”.
_ Page 8. Line 230. “… the PL spectra analysis was carried out”.
_ Page 8. Line 238. ¿“attribute” or “contribute”? in the sentence: “… perovskite film could attribute to recombination loss”.
Reviewer 3 Report
The authors investigated the effect of annealing temperature on electrical and morphological characteristic of E-beam evaporated TiO2 thin film as ETL to improve the performance of n-i-p planar PSC. The results are interesting and may merit publication in Micromachines although the device efficiency is a little low. Before recommending publication, I would like to ask the authors to carefully address the following questions:
1) In the conclusion, the authors stated that “It was revealed that the TiO2 film annealed at 480°C contained the lowest amount of surface defects and trap states which resulted in minimum recombination loss of photogenerated carries”. However, there is no characterization of defects (type, density, location) in the main text. It should be provided (such as SCLC, tDOS).
2) What is the strength of E-beam evaporated TiO2 compared to TiO2 prepared by other methods?
3) What is the innovation in this work compared to other reporters on E-beam evaporated TiO2?
4) Statistical data on device performance should be provided.
No
Round 2
Reviewer 1 Report
Thanks for your detailed reply. I don't have any further questions.
Reviewer 2 Report
The manuscript has been improved, but I have two comments:
1. Page 8, Lines 255-256. Authors claim: "..., with smaller grains resulting in a greater degree of confinement and a larger decrease in bandgap".
It is true that smaller grains results in a higher confinement degree. However, smaller grains results in a higher bandgap. Please correct or revise.
2. I did not understand the Response 6: Authors only said: "Please provide your response for Point 1. (in red)". Nevertheless, I have seen that additional information regarding to stability, PCE vs time and hysteresis have been added to the manuscript.
Reviewer 3 Report
The authors addressed all my points of criticism. I have no further comments.
No